# Health insurance, healthcare utilization and language use among populations who experience risk for tuberculosis, California 2014–2017

**Adam Readhead** [1,2]*, **Jennifer Flood**[1], **Pennan Barry**[1]

**1** Tuberculosis Control Branch, Division of Communicable Disease Control, Center for Infectious Diseases, California Department of Public Health, Richmond, California, United States of America, **2** Institute for Global Health Sciences, University of California San Francisco, San Francisco, California, United States of America

* adam.readhead@cdph.ca.gov

## Abstract

### Background

California tuberculosis (TB) prevention goals include testing more than ten million at-risk Californians and treating two million infected with tuberculosis. Adequate health insurance and robust healthcare utilization are crucial to meeting these goals, but information on these factors for populations that experience risk for TB is limited.

### Methods

We used data from the 2014–2017 California Health Interview Survey (n = 82,758), a population-based dual-frame telephone survey to calculate survey proportions and 95% confidence intervals (CI) stratified by country of birth, focusing on persons from countries of birth with the highest number of TB cases in California. Survey proportions for recent doctor's visit, overall health, smoking, and diabetes were age-adjusted.

### Results

Among 18–64 year-olds, 27% (CI: 25–30) of persons born in Mexico reported being uninsured in contrast with 3% (CI: 1–5) of persons born in India. Report of recent doctor's visit was highest among persons born in the Philippines, 84% (CI: 80–89) and lowest among Chinese-born persons, 70% (CI: 63–76). Persons born in Mexico were more likely to report community clinics as their usual source of care than persons born in China, Vietnam, or the Philippines. Poverty was highest among Mexican-born persons, 56% (CI: 54–58) and lowest among Indian-born persons, 9% (CI: 5–13). Of adults with a medical visit in a non-English language, 96% (CI: 96–97) were non-U.S.-born, but only 42% (CI: 40–44) of non-U.S.-born persons had a visit in a non-English language.

**Data Availability Statement:** Data cannot be shared publicly because of California Health Interview Survey protocol. Data are available from the University of California Los Angeles Center for

Health Policy Research at https://healthpolicy.ucla.edu/chis/data/Pages/GetCHISData.aspx for researchers who meet the criteria for access to confidential data.

**Funding:** The authors received no specific funding for this work.

**Competing interests:** The authors have declared that no competing interests exist.

## Discussion

Many, though not all, of the populations that experience risk for TB had health insurance and used healthcare. We found key differences in usual source of care and language use by country of birth which should be considered when planning outreach to specific providers, clinic systems, insurers and communities for TB prevention and case-finding.

## Introduction

Tuberculosis (TB) is the number one infectious disease killer worldwide causing 1.4 million deaths and ten million new cases of disease in 2019 [1]. In the United States (U.S.), substantial reductions in TB have been achieved though progress has stalled in recent years [2, 3].

California bears a disproportionate share of the TB burden in the U.S. In 2019, there were 2,113 TB cases in California, 24% of all cases nationally despite having 12% of the U.S. population [4, 5]. Country of birth is a major risk factor to TB disease. Of California TB cases, 84% occur among persons born outside the United States and rates of tuberculosis among non-U.S.-born Asians are 50 times higher than among U.S.-born whites [6]. In terms of burden of disease, persons born the United States, Mexico, the Philippines, Vietnam, and India accounted for 80% of persons with TB in California in 2019.

In California, more than 85% of cases result not from recent transmission of TB but from reactivation of latent tuberculosis infection (LTBI) [6, 7]. LTBI is usually acquired in countries of high TB incidence and can remain undiagnosed and untreated for years before progressing to active tuberculosis. To make progress against TB in California, the number of persons that are tested and treated for LTBI needs to be substantially increased. Recent mathematical models support the scale-up of targeted testing and treatment of non-U.S.-born persons for TB infection as important for the reduction of TB in California [8–11]. However, these TB prevention activities rely on access and use of healthcare and our knowledge of these attributes among non-U.S.-born persons is incomplete. In California and elsewhere, public health programs are engaging with community groups especially those focused on healthcare and representing affected communities but more information is needed about the demographics, healthcare utilization and potential barriers to care of populations that experience risk for TB [12–14]. Recent studies have highlighted disparities in healthcare utilization among minority groups but have not focused on non-U.S.-born persons who experience risk for TB [15, 16]. Understanding healthcare utilization of populations that experience risk is important for the planning of TB prevention activities.

One specific barrier to implementing TB prevention is that country of birth, the most important risk factor for TB, is often not captured in the electronic medical record (EMR) [17]. Without country of birth data, technology available in the EMR, such as prompts or reflex testing, cannot be used to promote TB testing among these groups that experience risk. Preferred language at medical visit, which is often populated in EMR data, may be useful as a proxy for country of birth.

Our aim was to describe the health insurance, healthcare utilization and language use of groups that experience risk for TB in California using a large-scale, population-based health survey to inform planning of TB prevention activities. As California transitions from TB control activities focused on finding and treating active TB disease to TB prevention activities requiring latent tuberculosis infection (LTBI) testing and treatment among populations that experience risk for TB, detailed knowledge about the population of the ten million persons in California born outside the U.S. is crucial.

## Methods

### Data source

Data were drawn from the California Health Interview survey (CHIS) for survey years 2014–2017 [18]. CHIS is an annual population-based telephone survey of California residents that employs both landline and cellphone random digit dialing [19]. Interviews were conducted in English, Spanish, Mandarin, Cantonese, Vietnamese, Korean and Tagalog. Details of the CHIS questionnaire have been published previously [20]. Data were based on respondent self-report and analysis was limited to CHIS adult survey which included persons 18 years and older (n = 82,758). There was insufficient data from the child and teen CHIS sub-surveys to calculate reliable estimates by country of birth.

### Definitions

We chose to examine health insurance, healthcare utilization and language use among persons born in Mexico, the United States, the Philippines, Vietnam, China, and India. When stratifying by country of birth, these six countries had the highest number of TB cases in California. Together they accounted for 80% of all cases in California during 2016–2017. Thus, for the purposes of this analysis, we defined populations that experience risk of TB as persons born in these six countries. Poverty was defined as less than 139% of the U.S. federal poverty level. The respondent's percentage of federal poverty level was calculated using household income earned in the U.S. and household size. Educational attainment was estimated among persons 25 years old and older. Language at medical visit was defined as the language spoken by doctor at last medical visit excluding those who had difficulty communicating with their doctor (<5%) or without a medical visit in last two years. Respondents who did not answer this question but had a medical visit in the last two years and conducted their survey in English were categorized as speaking English at the medical visit. High deductible plans were defined as those plans with an annual deductible of more than $1,000. Delayed or forgone medical care and delayed or forgone medicine were defined as events occurring in the last 12 months; these questions were asked of persons with health insurance in the last 12 months.

Treated unfairly when getting medical care was defined as being treated unfairly sometimes or often when getting medical care over lifetime. Racial or ethnic discrimination was defined as responding yes to the question "was there ever a time when you would have gotten better medical care if you had belonged to a different race or ethnic group?" Usual source of care was condensed from seven categories to three as follows: "doctor's office", "health maintenance organization (HMO) or Kaiser" were grouped as "doctor's office"; "community clinic or hospital" and "government clinic or hospital" were grouped as "community/government clinic/hospital"; categories of "no usual place", "emergency room", "urgent care", "other place", and "no one place" were grouped into a single category "no usual source." Data on unfair treatment when getting medical care, perceived racial of ethnic discrimination, and diabetes (non-gestational) were limited to 2015–2017 because these questions were not included in the 2014 survey. Smoking and diabetes were considered because they are risk factors for progression of latent TB infection to TB disease [21, 22].

### Statistical analysis

We calculated survey proportions and 95% confidence intervals (CI) stratified by country of birth. We adjusted the following variables by age: doctor's visit in the last 12 months, overall health, smoking and diabetes. We used the 2017 U.S. Census annual estimate of population for California as the standard population [23]. Survey proportions based on less than three

respondents, or 500 weighted respondents were suppressed per California Department of Public Health guidelines for CHIS use. Analysis was done in SAS 9.4. The California Health and Human Services Agency's Committee for the Protection of Human Subjects (Federal-wide Assurance #00000681) determined that this project was not human subjects research and did not require ethics committee approval or informed consent to be conducted.

## Results

### Demographic and socioeconomic factors

Demographic and socioeconomic factors of populations that experience risk for TB including sex, age, length of residency in the U.S., education, poverty, and English language proficiency differed by country of birth (Table 1). The proportion of females by country of birth ranged from 60% (CI: 52–68) among Chinese-born persons to 43% (CI: 34–51) among Indian-born persons. The proportion of 18–29 year-olds among U.S.-born was 27% (CI: 27–27) with far lower proportions among persons born outside the U.S., ranging from 8% (CI: 4–11) among those born in Vietnam to 19% (CI: 14–24) among those born in China. Close to three-quarters of persons born in Vietnam or Mexico reported residing in the U.S. for 16 years or more compared with two-thirds of persons born in the Philippines and half of persons born in China or India.

There were substantial differences in educational attainment by country of birth. Of persons born in Vietnam, 55% (CI: 48–63) had a high school education or less. Among persons born in Mexico, that estimate was 83% (CI: 81–85). In comparison, the proportion with a high school education or less was 13% (CI: 8–19) among persons born in the Philippines. Poverty followed similar patterns to educational attainment, with some notable differences. Among persons born in Mexico or Vietnam, 56% (CI: 54–58) and 47% (CI: 37–55) respectively lived in poverty. In contrast, 19% (CI: 18–20) and 9% (CI: 5–13) of persons born in the U.S. or India lived in poverty. Also, most non-U.S.-born persons had resided in the U.S. for 16 or more years, across all countries of birth.

### Health insurance

Patterns of healthcare insurance were also notably different (Table 2). The proportion of 18–64 year-olds who were uninsured ranged from 27% (CI: 25–30) among persons born in Mexico to 3% (CI: 1–7) among persons born in India. Among persons 65 years old and older, the proportion who reported both Medicare and Medi-Cal coverage ranged from 14% (CI: 12–16) among U.S.-born to 72% (CI: 59–84) among persons born in Vietnam. More than half of Indian-born persons, 51% (CI: 39–62), reported high deductible health insurance plans in contrast to a third of U.S.-born persons who reported the same, 33% (CI: 32–34).

### Healthcare utilization and barriers to care

The proportion of persons reporting visiting a doctor in the last 12 months differed by country of birth, even after age adjustment (Table 2). Report of age-adjusted recent doctor's visit was highest among persons born in the Philippines, 84% (CI: 80–89) and lowest among Chinese-born persons, 70% (CI: 63–76). Age-adjusted recent doctor's visit among persons born in the U.S. was 83% (CI: 83–84).

There was also a substantial difference in usual source of care by country of birth (Table 2). The proportion reporting a doctor's office as usual source of care was highest among Vietnamese-born persons at 73% (CI: 66–81) and lowest among Mexican-born persons, 31% (CI: 24–34). The proportion reporting no usual source of care was highest among Mexican-born persons at 28% (CI: 26–30) and lowest among Indian-born persons at 11% (CI: 6–15).

**Table 1. TB Burden and demographic characteristics by country of birth, California 2014–2017.**

| | | Philippines | Vietnam | India | China | Mexico | United States |
|---|---|---|---|---|---|---|---|
| Incidence of TB disease per 100,000 person-years | | 45 | 41 | 25 | 23 | 10 | 1 |
| Proportion of overall TB cases | | 18 | 10 | 5 | 7 | 21 | 19 |
| | | % (95% CI) | % (95% CI) | % (95% CI) | % (95% CI) | % (95% CI) | % (95% CI) |
| Sex | | | | | | | |
| | Female | 58 (51–64) | 54 (45–62) | 43 (34–51) | 60 (52–68) | 51 (49–52) | 51 (50–51) |
| | Male | 42 (36–49) | 46 (38–55) | 57 (49–66) | 40 (32–48) | 49 (48–51) | 49 (49–50) |
| Age | | | | | | | |
| | 18–29 | 17 (11–22) | 8 (4–11) | 19 (12–26) | 19 (14–24) | 11 (9–12) | 27 (27–27) |
| | 30–39 | 12 (8–16) | 14 (9–20) | 29 (21–37) | 19 (13–24) | 23 (21–25) | 17 (16–17) |
| | 40–49 | 22 (16–28) | 28 (20–36) | 25 (18–32) | 24 (18–29) | 29 (27–31) | 13 (13–14) |
| | 50–59 | 18 (14–23) | 19 (13–24) | 12 (6–18) | 13 (7–19) | 20 (18–22) | 16 (15–16) |
| | 60–69 | 17 (13–22) | 18 (11–25) | 9 (5–12) | 13 (8–17) | 12 (10–13) | 14 (14–15) |
| | 70–79 | 10 (7–14) | 11 (6–15) | 5 (2–9)* | 8 (4–13) | 4 (3–5) | 8 (8–9) |
| | 80+ | 4 (2–6) | 3 (1–5)* | 1 (0–3)* | 5 (2–7) | 1 (1–2) | 5 (5–5) |
| Years in the U.S. | | | | | | | |
| | 0–5 | 11 (6–15) | 7 (3–11) | 21 (14–29) | 19 (14–25) | 4 (3–5) | |
| | 6–10 | 12 (8–16) | 12 (6–19) | 13 (7–19) | 16 (10–21) | 8 (7–10) | |
| | 11–15 | 12 (8–17) | 7 (4–11) | 16 (10–23) | 12 (8–16) | 14 (12–15) | |
| | 16+ | 65 (59–70) | 73 (64–81) | 49 (41–57) | 53 (46–60) | 74 (72–76) | |
| Educational Attainment (of 25 years old and older) | | | | | | | |
| | High school/HS equivalent or less | 13 (8–19) | 55 (48–63) | 4 (1–7)* | 30 (24–35) | 83 (81–85) | 27 (26–28) |
| | Some College/Vocational School/AA or AS | 23 (17–29) | 10 (7–14) | 5 (1–9)* | 7 (4–10) | 9 (8–10) | 28 (27–29) |
| | BA/BS degree or higher | 64 (57–71) | 34 (25–43) | 91 (87–95) | 64 (57–70) | 8 (7–9) | 45 (44–46) |
| Poverty Level | | | | | | | |
| | 0–138% FPL | 26 (20–32) | 47 (39–55) | 9 (5–13) | 29 (23–35) | 56 (54–58) | 19 (18–20) |
| | 139%-249% FPL | 18 (14–23) | 16 (11–21) | 8 (4–13) | 14 (9–18) | 26 (24–28) | 16 (15–17) |
| | 250%-399% FPL | 18 (13–22) | 10 (6–14) | 15 (8–21) | 15 (10–20) | 11 (10–13) | 18 (17–19) |
| | 400%+ FPL | 38 (32–45) | 27 (19–35) | 68 (60–75) | 42 (35–49) | 7 (5–8) | 47 (46–48) |
| Language spoken at home | | | | | | | |
| | English only | 22 (17–28) | 7 (1–12)* | 9 (5–13) | 7 (4–10) | 2 (1–3) | |
| | Language of country of birth and English | 59 (54–65) | 30 (22–37) | 70 (62–78) | 34 (28–41) | 47 (44–49) | |
| | Other including multiple non-English languages | 9 (6–13) | 11 (6–16) | 7 (3–11)* | 12 (7–18) | 2 (1–2) | |
| | Language of country of birth only | 9 (6–12) | 53 (45–61) | 14 (9–19) | 46 (39–53) | 50 (47–52) | |
| English language proficiency | | | | | | | |
| | Speaks English only | 22 (17–28) | 7 (1–12)* | 9 (5–13) | 7 (4–10) | 2 (1–3) | 77 (76–78) |
| | Very well/Well | 71 (64–77) | 40 (32–48) | 89 (85–94) | 54 (47–60) | 31 (29–33) | 23 (22–24) |
| | Not well/Not at all | 7 (3–12)* | 53 (45–61) | 1 (0–3)* | 39 (33–45) | 67 (65–69) | 1 (0–1) |

Source: Data on TB incidence and burden are from the Tuberculosis Control Branch, California Department of Public Health 2017; All other data from California Health Interview Survey, 2014–2017.

Notes

* Statistically unstable—Coefficient of Variation > 0.3.

The proportion of persons experiencing barriers to care was low. Whereas 11% (CI: 11–12) of U.S.-born persons had delayed or forgone medicine, 4% (CI: 2–7) of Chinese-born persons had done the same. Similarly, 14% (14%-15%) of U.S.-born persons had delayed or forgone medical care, whereas 7% (4–10) of Chinese-born persons had done the same.

Few people reported racial discrimination and unfair treatment, and, in the case of discrimination, variable by country of birth though these estimates were statistically unstable. Of U.S.-born persons, 10% (CI: 9–11) reported being treated unfairly sometimes or often when getting medical care. Of persons born Mexico, 14% (CI: 11–16) reported the same. In contrast, of persons born in China 7% (CI: 3–11) reported this. Less than 10% reported having a hard time understanding the doctor at last visit with the highest proportion among Mexican-born persons at 7% (CI: 5–8) and Chinese-born persons 6% (CI: 2–9). For comparison, the estimate was 3% (CI: 2–3) for U.S.-born persons.

## Overall health and risk factors for TB reactivation

Age-adjusted overall health differed widely by country of birth (Table 2). Among U.S.-born, 54% (CI: 52–55) reported very good or excellent health. Lower proportions of good or excellent health were reported by persons born in Mexico or in Vietnam: 26% (CI: 24–28) and 29% (CI: 21–38). The highest proportion with very good or excellent health was among persons born in India at 62% (CI: 52–71).

**Table 2. Healthcare access and utilization by country of birth, California 2014–2017.**

| | | Philippines | Vietnam | China | India | Mexico | United States |
|---|---|---|---|---|---|---|---|
| Health insurance (18–64) | | | | | | | |
| | Employment-Based /Privately Purchased | 63 (56–71) | 54 (44–64) | 70 (64–76) | 88 (82–93) | 32 (30–34) | 65 (64–66) |
| | Medi-Cal/ Medicare/ Other Public | 24 (17–32) | 34 (24–44) | 19 (13–24) | 10 (4–15) | 41 (38–44) | 26 (25–27) |
| | Uninsured | 12 (8–17) | 12 (6–17) | 11 (7–15) | 3 (1–5)* | 27 (25–30) | 9 (8–9) |
| Health insurance (65+) | | | | | | | |
| | Medicare + Medi-Cal | 37 (27–47) | 72 (59–84) | 37 (22–53) | 17 (0–38)* | 50 (44–56) | 14 (12–16) |
| | Medicare + Other | 48 (37–59) | 20 (10–30) | 53 (37–69) | 60 (38–81) | 32 (26–38) | 75 (73–77) |
| | Medicare Only, Other Only and Uninsured | 15 (6–23)* | 8 (0–17)* | 10 (3–17)* | 23 (5–41)* | 18 (13–23) | 11 (10–12) |
| High deductible (of employer/private insured) | | 38 (30–47) | 43 (29–58) | 46 (35–58) | 51 (39–62) | 39 (35–42) | 33 (32–34) |
| Usual source of care | | | | | | | |
| | Doctor's Office | 55 (48–63) | 73 (66–81) | 55 (49–62) | 61 (53–69) | 31 (28–34) | 65 (64–67) |
| | Community/government clinic or hospital | 29 (23–36) | 7 (4–10) | 27 (20–33) | 28 (20–36) | 41 (38–45) | 19 (18–20) |
| | No Usual Source | 15 (11–19) | 19 (13–26) | 18 (13–23) | 11 (6–15) | 28 (26–30) | 15 (14–16) |
| Visited doctor in last 12 months § | | 84 (80–89) | 76 (67–85) | 70 (63–76) | 82 (76–88) | 73 (71–75) | 83 (83–84) |
| Hard time understanding doctor at last visit | | 1 (0–3)* | 6 (2–10)* | 6 (3–9) | ‡ | 7 (5–8) | 3 (2–3) |
| Delayed or forgone prescription drugs | | 7 (4–11) | 4 (1–7)* | 4 (2–7) | 6 (0–13)* | 9 (8–11) | 11 (11–12) |
| Delayed or forgone medical care | | 7 (4–10) | 7 (2–12)* | 7 (4–10) | 8 (4–12) | 10 (9–11) | 14 (14–15) |
| Treated unfairly when getting medical care† | | 9 (5–13) | 11 (5–16) | 7 (3–11) | 11 (3–19)* | 14 (11–17) | 10 (10–11) |
| Racial or ethnic discrimination† | | 2 (0–4)* | 6 (2–10)* | 8 (2–13)* | 7 (0–14)* | 6 (5–8) | 4 (4–5) |
| Excellent/very good overall health § | | 52 (46–58) | 29 (21–38) | 46 (39–53) | 62 (52–71) | 26 (24–28) | 54 (52–55) |
| Smoking § | | | | | | | |
| | Currently Smokes | 8 (5–12) | 11 (6–17) | 7 (3–10) | 4 (1–7) | 10 (8–11) | 13 (13–14) |
| | Quit Smoking | 19 (14–23) | 8 (4–12) | 8 (5–12) | 10 (6–15) | 18 (16–20) | 24 (23–25) |
| | Never Smoked Regularly | 73 (67–79) | 81 (74–87) | 85 (80–90) | 86 (80–91) | 73 (70–75) | 63 (62–64) |
| Ever told had diabetes (excludes gestational)†§ | | 15 (9–20) | 7 (4–10) | 7 (3–11)* | 9 (4–14) | 17 (15–19) | 8 (8–9) |

Source: California Health Interview Survey, 2014–2017.

Notes

* Statistically unstable—coefficient of variation > 0.3

§ Age adjusted

‡ Estimate suppressed per California Health Interview Survey guidelines

† Data not available for 2014. Estimate calculated using 2015–2017 data.

**Table 3. Language at medical visit by nativity, California 2014–2017.**

|  | Non-U.S.-born | U.S.-born |  |
|---|---|---|---|
| Non-English | 3,509,456 | 128,524 | 3,637,980 |
| English | 4,839,626 | 17,281,183 | 22,120,809 |
|  | 8,349,082 | 17,409,707 | 25,758,789 |
|  | Sensitivity | 42% |  |
|  | Specificity | 99% |  |
|  | Positive predictive value | 96% |  |
|  | Negative predictive value | 78% |  |

Source: California Health Interview Survey, 2014–2017.

Note: Weighted totals are presented.

Current smoking was highest among persons born in Vietnam 11% (CI: 6–17) though the confidence interval was wide. Persons born in Mexico, or the Philippines reported high proportions of former smoking, 18% (CI: 16–20) and 19% (CI: 14–23) respectively. Of persons born in the U.S., 24% (CI: 23–25) reported former smoking. Former smoking was lower among persons born in Vietnam, China, or India, 8% (CI: 4–12), 8% (CI: 5–12) and 10% (CI: 6–15) respectively.

Of persons born in Mexico or the Philippines, 17% (CI: 14–19) and 17% (CI: 12–23) reported diabetes respectively (age-adjusted). Diabetes proportions were lower among persons born in Vietnam, China, and India, similar to the level observed among the U.S.-born, 8% (CI: 8–9).

## Language at medical visit and at home

Non-English language use at medical visit was indicative of birth outside the U.S.; of adults with a medical visit in a non-English language, 96% (CI: 96–97) were non-U.S.-born (Table 3). However, of non-U.S.-born, 42% (CI: 40–44) had medical visits in a non-English language. As a proxy for nativity, language at medical visit had a low sensitivity (42%) and a high specificity (99%) (Table 3).

Further stratifying by language used, we see that non-English language use was correlated closely with non-U.S. birth across languages (Table 4). Of persons with visits in Tagalog, 96%

**Table 4. Proportion country of birth by language spoken in medical visit, California 2014–2017.**

| Language | United States % (95% CI) | Philippines % (95% CI) | Vietnam % (95% CI) | China % (95% CI) | India % (95% CI) | Korea % (95% CI) | Mexico % (95% CI) | Other Country |
|---|---|---|---|---|---|---|---|---|
| English | 78 (78–79) | 3 (2–3) | 1 (1–1) | 1 (1–1) | 1 (1–2) | 0 (0–1) | 6 (6–7) | 9 (8–10) |
| Tagalog |  | 96 (88–100) |  |  |  |  |  |  |
| Vietnamese | 2 (0–4)* |  | 96 (92–100) |  |  |  |  |  |
| Mandarin | 4 (0–8)* |  | 2 (0–3)** | 70 (59–80) |  |  |  | 22 (13–32) |
| Cantonese |  |  | 8 (0–26)** | 85 (66–100) |  |  |  | 6 (0–14)* |
| Asian Indian Languages |  |  |  |  | 80 (49–100) |  |  | 14 (0–42)* |
| Korean |  |  |  |  |  | 99 (96–100) |  |  |
| Spanish | 4 (3–5) |  |  |  |  |  | 78 (75–81) | 19 (16–21) |

Source: California Health Interview Survey, 2014–2017

Notes

* Statistically unstable—Coefficient of Variation > 0.3.

(CI: 88–100) were born in the Philippines. Of persons with visits in Vietnamese, 96% (CI: 92–100) were born in Vietnam. Proportion non-U.S.-born for visits in Mandarin, Cantonese or an Asian Indian language were similar. Of persons with visits in Spanish, 78% (CI: 75–81) were born in Mexico and 19% (CI: 12–21) elsewhere outside the U.S.

The language used at medical visit varied widely by country of birth (S1 Table). Among persons born in Mexico or Vietnam, 38% (CI: 36–41) and 41% (CI: 31–50) had visits in English; similar results were found for persons born in China. In contrast, among persons born in India or the Philippines, 96% (CI: 94–99) and 91% (CI: 87–95) had a visit in English. Of U.S.-born, 0.7% (CI: 0.5–0.9) had visits in a non-English language.

Of adults who did not speak English at home, 89% (CI: 88–90) were non-U.S.-born. Among non-U.S.-born, 43% (CI: 41–44) did not speak English at home. The proportion of persons residing in households in which no English was spoken ranged widely by country of birth (Table 1). Of persons born in the Philippines, 9% (CI: 6–12) spoke only Tagalog at home and, of persons born in India, 14% (CI: 9–19) spoke only an Asian Indian language at home. In contrast, 53% (CI: 46–61) of persons born in Vietnam spoke only Vietnamese at home and 50% (CI: 47–52) of persons born in Mexico spoke only Spanish at home. English proficiency estimates followed similar patterns.

## Discussion

In our analysis of a representative self-reported health survey focusing on non-U.S.-born populations that experience risk for tuberculosis, we have identified five findings that could help inform future TB prevention activities in California and across the U.S. These findings present opportunities and signal potential pitfalls for planning outreach to specific providers, clinic systems, insurers, and communities.

First, most persons who experience risk for TB had health insurance and were engaged with medical care. Analyses of health insurance coverage among non-U.S.-born persons using a variety of national datasets have reported similar results [24–26]. Barriers to healthcare access and utilization, such as cost and racial/ethnic discrimination, affected less than one in ten persons. Racial or ethnic discrimination has been shown to reduce access care and filling of prescriptions which would hamper TB prevention activities [27, 28]. With the notable exceptions of persons experiencing homelessness and one quarter of persons born in Mexico who are uninsured, the hurdle of getting populations that experience risk into care has largely been met. Future efforts should focus on encouraging patients who experience risk to ask for TB testing and treatment and aiding providers in their efforts to test and treat persons already in care. Public awareness campaigns could consider intervening with patients at or around the point of care.

Second, populations that experience risk had notably different usual sources of care. In agreement with previous studies, persons born in Mexico were more likely to use community or government facilities or to have no usual source of care than other populations that experience risk [26]. Although community and government providers are natural partners for public health programs, persons born in countries with the highest TB rates like Vietnam and the Philippines were less likely to use these providers. While more than a third of Vietnamese-born 18–64 year-olds and one-fifth of those over 65 years old had public insurance, less than one in ten used government or community facilities. Consistent with other reports, this finding suggests that efforts to reach populations that experience risk should not focus solely on public healthcare systems and publicly funded clinics but must engage both public and private healthcare systems [25]. In addition, type of health insurance alone cannot predict the usual source of care in these populations that experience risk. As is shown with insurance and access

patterns of the Vietnamese-born above, persons with government insurance like Medi-Cal do not necessarily use government or community clinics. This is important to keep in mind when considering type of health insurance as a proxy for usual source of care.

Third, risk factors for the progression of latent TB infection to TB disease varied widely among the populations surveyed. Diabetes and smoking are associated with increased risk of progression of latent TB infection to active TB disease [21, 22]. Overall health may be associated with TB disease [29]. Self-reported diabetes among Philippines-born and Mexican-born persons was more than double the rates for persons born in other countries including the U.S., similar to prior work [30, 31]. Likewise, former smoking was twice as high among Philippines-born and Mexican-born persons than among persons born in China, India or Vietnam as has been previously documented [32, 33]. The prevalence of these risk factors among populations who already experience risk for TB provides opportunities for engagement with providers who may not currently think about TB even when seeing persons who experience risk for the disease. One engagement strategy could be using a common comorbidity such as diabetes as an entry point for provider education about TB. It also highlights an opportunity to collaborate with organizations which focus on these comorbidities. The dual intervention of smoking cessation and latent TB infection targeted testing and treatment could have a powerful effect by averting costly and deadly disease in key groups.

Fourth, there were prominent differences in the demography of populations that experience risk that could inform the way in which these populations are engaged. Persons born in Mexico or Vietnam were more likely to be long-term residents of the U.S., have lower education attainment, higher poverty levels and lower self-reported overall health in comparison to persons born in the Philippines, China, or India. Roughly, one-fifth of U.S.-born persons lived in poverty whereas one quarter to one half of persons born in Mexico, Vietnam, China, or the Philippines were impoverished. Less than one in 10 persons born in India lived in poverty. In addition, persons born in Mexico, Vietnam or China were more likely not to speak English at their medical visit or at home than persons born in India or the Philippines. This information, especially language use and educational attainment, has immediate, practical use in tailoring public-facing campaigns, specifically the languages and reading levels in which educational materials are offered.

Fifth, we investigated whether language at medical visit could be used to identify persons born outside the U.S. We found that persons who used a language other than English at their medical visit were highly likely to be born outside the U.S. Thus, preferred language may be suitable as a starting point for identifying high-risk subgroups. However, non-English language at medical visit identified only half of all non-U.S.-born persons. EMRs should be modified to accommodate country of birth data, which is commonly missing in these records, and workflows should be adjusted to collect these data. Without country of birth data, a large portion of the population who experience risk could be missed. Even with system modifications, previous reports have noted that collection of these data can be difficult to navigate with patients [34]. Still, use of preferred language to identify patients who experience risk could reduce a major barrier to implementing TB prevention activities in large health systems where country of birth, the most important risk factor for latent TB infection, is often not captured.

Language used at medical visit and at home were roughly concordant which provides an opportunity to use language at home when language at medical visit is not available. Non-English speaking persons could be at increased risk for TB compared with other persons from the same country because previous studies have shown non-U.S.-born persons with low educational attainment are at increased risk of TB and, from our analysis, we see non-U.S.-born persons with low educational attainment are more likely not to speak English [35, 36].

This analysis has several limitations. Recent shifts in employment and health insurance coverage due to the COVID-19 pandemic may reduce the applicability of this work. Updates to

these analyses will be necessary. These results are most useful if, within country of birth strata, persons with LTBI have similar health care access and utilization patterns as those without LTBI. A recent national analysis suggests that this is the case [25]. While recent analyses have detailed heterogeneity in health insurance coverage by immigration legal status, similar analyses were not possible here because these data were not collected [26]. Also, data useful to describing TB risk such as travel to country of origin and household crowding were not available. The survey was conducted by phone and may under-represent persons who are less likely to answer the phone, such as persons experiencing homelessness, young adults, and non-English speakers. Furthermore, it relies on participant self-report which may underestimate key determinants of TB risk, such as poverty and country of origin, because of social desirability bias. Because this analysis focuses on non-U.S.-born persons, and persons experiencing homelessness maybe under-represented, it would be helpful to know the proportion of non-U.S.-born persons who experience homelessness in California. Unfortunately, there is no state-wide estimate for non-U.S.-born homelessness, but a recent study found the proportion of lifetime homelessness among non-U.S.-born persons was 1% [37]. However, from California TB data, homelessness was less common among non-U.S.-born persons with TB at 3.6%. Among U.S.-born cases that proportion was 15.2%. Language used by patient at medical visit was not assessed in the survey; language used by doctor at medical visit was used instead. Data on providers serving these populations at-risk for TB was not available limiting the ability to identify these providers for potential awareness campaigns.

## Conclusion

Our analysis points out several important differences by country of birth in demography, healthcare utilization, and language use among populations that experience risk for TB. These differences present opportunities and signal potential pitfalls for planning outreach to specific communities that experience risk for TB and the providers, clinic systems, health insurers and health insurance purchasers who serve them. These data could be used to inform public awareness campaigns, provider education, academic detailing, and community outreach efforts. We envision these efforts will be multi-faceted, multi-sectorial, and adapted to specific communities and providers and that they include input from populations that experience risk for TB. While the information here most directly supports this tailored approach in California, we believe that similarly crafted approaches would help TB prevention efforts in other states across the U.S. as well.

## Supporting information

**S1 Table. Proportion language spoken in medical visit by country of birth, California 2014–2017.**
(DOCX)

## Acknowledgments

The authors would like to acknowledge Alex Golden, David Dauphine, Jennifer Rico at the California Department of Public Health, and staff at the UCLA Center for Health Policy Research for their assistance in this project.

## Author Contributions

**Conceptualization:** Adam Readhead, Pennan Barry.

**Data curation:** Adam Readhead.

**Formal analysis:** Adam Readhead.

**Methodology:** Adam Readhead, Pennan Barry.

**Supervision:** Jennifer Flood.

**Writing – original draft:** Adam Readhead, Pennan Barry.

**Writing – review & editing:** Adam Readhead, Jennifer Flood, Pennan Barry.

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
