## [Decision Letter · Decision Letter 0]

24 May 2021

PONE-D-21-08860

Navigating toward TB elimination: the California perspective.

PLOS ONE

Dear Dr. Readhead,

Thank you for submitting your manuscript to PLOS ONE. After careful consideration, we feel that it has merit but does not fully meet PLOS ONE’s publication criteria as it currently stands. Therefore, we invite you to submit a revised version of the manuscript that addresses the points raised during the review process.

Please submit your revised manuscript by July 2nd, 2021. If you will need more time than this to complete your revisions, please reply to this message or contact the journal office at plosone@plos.org. Please include the following items when submitting your revised manuscript:

We look forward to receiving your revised manuscript.

Kind regards,

Rubeena Zakar, Ph.D

Academic Editor

PLOS ONE

Journal Requirements:

2. In your Methods section, please provide additional information about the analysis performed, for example how many participants were included.

3. Please consider modifying your title to ensure that it is specific and descriptive.

4a) If there are ethical or legal restrictions on sharing a de-identified data set, please explain them in detail (e.g., data contain potentially identifying or sensitive patient information) and who has imposed them (e.g., an ethics committee). Please also provide contact information for a data access committee, ethics committee, or other institutional body to which data requests may be sent.

4b) If there are no restrictions, please upload the minimal anonymized data set necessary to replicate your study findings as either Supporting Information files or to a stable, public repository and provide us with the relevant URLs, DOIs, or accession numbers. Please see http://www.bmj.com/content/340/bmj.c181.long for guidelines on how to de-identify and prepare clinical data for publication. For a list of acceptable repositories, please see http://journals.plos.org/plosone/s/data-availability#loc-recommended-repositories.

6. Please include your tables as part of your main manuscript and remove the individual files. Please note that supplementary tables (should remain/ be uploaded) as separate "supporting information" files

Reviewers' comments:

Reviewer's Responses to Questions

**Comments to the Author**

1. Is the manuscript technically sound, and do the data support the conclusions?

Reviewer #1: Yes

Reviewer #2: Yes

Reviewer #3: Partly

2. Has the statistical analysis been performed appropriately and rigorously? 

Reviewer #1: Yes

Reviewer #2: Yes

Reviewer #3: I Don't Know

3. Have the authors made all data underlying the findings in their manuscript fully available?

Reviewer #1: Yes

Reviewer #2: Yes

Reviewer #3: Yes

4. Is the manuscript presented in an intelligible fashion and written in standard English?

Reviewer #1: Yes

Reviewer #2: Yes

Reviewer #3: Yes

5. Review Comments to the Author

Reviewer #1: It would be nice to have input from your populations of interest (foreign born people) – either to put in this paper, or as future steps as you engage with providers and patients at risk. Also, what is currently being done in terms of outreach to specific communities? What have other states done? Nice to have more CA specific background stats on TB and more specifically immigrant populations – what makes CA different/better/worse than other states?

Not sure why looking at attitudes/feelings around discrimination – while interesting, they don’t really add to the efforts/next steps around TB prevention (or this wasn’t made clear in the paper)

Line 230 – not clear what % of homeless are foreign born. They couldn’t be surveyed, so how do we know they are not a large %?

Need to clearly elucidate next steps with regard to improving EMR documentation of country of birth

Is there anything that can be included in the next statewide survey to help this TB campaign?

Good point about leveraging comorbidity work (smoking/diabetes).

It seems like different countries at birth (e.g. Mexico) have different needs – maybe focus on some of those. County of birth being Mexico has the largest proportion of TB cases (in Table 1, 21), and in lines 230-231 you state ¼ are uninsured – seems like a good part to focus prevention efforts on. This pop also has a high percentage of diabetes.

Focus on provider education in terms of increased testing for TB in this population/routine testing (not patient – in line 232). And who are these providers caring for foreign born populations? (do they differ than other providers?)

What about the following factors in describing your pop of interest: legal status (esp for people interested in obtaining legal status – might be less likely to utilize any services out of fear of jeopardizing their future greencards), travel back to country of origin (perhaps this is where they get TB, thru travel, and not just coming into the US with it), age at immigration to US, household size/make up of extended family (and if they have travel back to at-risk countries, or issues of household crowding making it easier to spread TB).

Would be nice know breakdown of active TB cases (foreign born vs. US born), does this follow the same patterns as described in this paper?

Table 1 – more “TB Burden” to be at top of table, not at bottom

Table 4 – include percentages

Reviewer #2: Overall, I found this an interesting analysis of the demographic, health, and healthcare seeking characteristics of the non-US born population in California. While this population is certainly important to identify, describe, and understand for TB control, the paper lacks a clearly presented conceptual framework that describes how these demographic and clinical characteristics: 1) impact TB care and outcomes; and 2) are unique to TB. The non-USB population is at risk for a variety of conditions (Hepatitis, COViD, e.g.) making it important to clearly describe the context for focus on non-USB more explicitly in the introduction. A conceptual framework for why each of these data fields from CHIS were chose would also be helpful, especially for readers not as familiar with the domestic TB literature.

Please specify in Methods why you focus on 5 countries named (assuming based on country of origin for most TB cases in CA?).

Suggest a citation for this statement as it forms a key underlying premise for the purpose of the paper (ln 67)” One specific barrier to implementing TB prevention in large health systems is that country of 68 birth, the most important risk factor for TB, is often not captured in the electronic medical 69 record (EMR).”

This statement does not seem to make sense (ln 277-278) “Our results show that non-English language at medical visit was an excellent proxy for non-U.S. 278 birth, but identified only half of all non-U.S.-born persons.” How is a proxy excellent if it only identifies half of the eligible population? Please refer to the 2-way table you present in results as to how you would like the reader to interpret this statement. Are you say that the positive predictive value or specificity is high but sensitivity is low? If so, please state as such.

Please describe important limitations of using CHIS data, ie, sampling bias based on phone and self-response and how this may actually underestimate the prevalence of some of those key sociodemographic determinants that affect TB care.

Table 3 is challenging to read and interpret. Suggest revising.

You may want to consider using the first paragraph of the Discussion as the last paragraph of the Introduction, as it is the first place that seems to clearly articulate the purpose for the analysis presented.

Reviewer #3: This is an interesting paper. It has the potential benefits in the field of TB in the USA. But to let it be strong enough, some clarifications are needed. Below are some suggestions which could help improve the paper if the authors found them relevant.

- I would suggest to further explain the methodology. I’m a little bit concerned with the methodology section as presented. The methods section should be clearly and unambiguously stated. I find that there are many references that are indicated and that unfortunately hide a lot of details. For example, the methodology does not show the countries or let's say the different nationalities (India, Mexico, USA, Philippine, Vietnam, etc.) mentioned in the results section. Is there any link or relationship between individuals regarding their nationalities and the risk to get TB? Also, what is the threshold and the criteria of poverty according to the place of birth? According to the fact that poverty in the US could be completely different of the one in Philippine or somewhere else…

- HMO. I don’t know what this acronym means! Please put the definition in bracket (p.99)

- In most of the items presented in the results section, it is not easy to perceive the link between the different factors (diabetes, smoking, good or bad health, etc.) mentioned and their relationship with TB. The authors should discuss this further.

- The discussion section, as presented, in my opinion is more a part where the authors interpret their results and not discussing them. The discussion, as its name indicates, should be the place where the results, while being interpreted, are also confronted with the studies of other authors and where the points of view are clearly confirmed, refuted or nuanced. And this should be felt in the richness of the references mentioned in the said section. But, as we can see here, the authors have mentioned only 2 references in this whole long discussion... And besides, the article is not rich enough in terms of bibliographic references. So, I suggest to enrich it further, if only with a little more reading...

- Where is the conclusion section? And what are the main conclusions of the data obtained from the California Health Interview survey (CHIS)?

- What could be the concrete added value of this article? What contribution and especially how can the results obtained contribute to the elimination of TB in the whole USA in general or at least in California more specifically? This is not clear in the paper.

6. PLOS authors have the option to publish the peer review history of their article (what does this mean?). If published, this will include your full peer review and any attached files.

Reviewer #1: **Yes: **Kelly Kathleen McCabe

Reviewer #2: No

Reviewer #3: No

---

## [Author Response · Author response to Decision Letter 0]

31 Aug 2021

Response to editor’s and reviewers’ comments:

Editor

 DONE.

2. In your Methods section, please provide additional information about the analysis performed, for example how many participants were included.

DONE. Methods now include the line: “Data were based on respondent self-report and analysis was limited to persons 18 years and older (n=82,758).”

3. Please consider modifying your title to ensure that it is specific and descriptive.

DONE. The manuscript is now titled “Demographics and healthcare utilization of populations at risk for tuberculosis, California 2014-2017.” 

4a) If there are ethical or legal restrictions on sharing a de-identified data set, please explain them in detail (e.g., data contain potentially identifying or sensitive patient information) and who has imposed them (e.g., an ethics committee). Please also provide contact information for a data access committee, ethics committee, or other institutional body to which data requests may be sent.

DONE. Addressed in cover letter.

DONE. Ethics statement now included in methods section and deleted from other sections.

6. Please include your tables as part of your main manuscript and remove the individual files. Please note that supplementary tables (should remain/ be uploaded) as separate "supporting information" files

DONE. Tables are now included as part of main manuscript file.

Reviewers' comments

Reviewer #1: It would be nice to have input from your populations of interest (foreign born people) – either to put in this paper, or as future steps as you engage with providers and patients at risk. 

Revised to address comment: “We envision these efforts will be multi-faceted, multi-sectorial, and adapted to specific communities and providers and that they include input from populations at risk for TB”

Also, what is currently being done in terms of outreach to specific communities? What have other states done? 

Added the following text to the introduction: “In California and elsewhere, public health programs are engaging with community groups especially those focused on healthcare and representing affected communities…”

Nice to have more CA specific background stats on TB and more specifically immigrant populations – what makes CA different/better/worse than other states?

Added CA background statistics to the introduction: “California bears a disproportionate share of the TB burden in the U.S. In 2019, there were 2,113 TB cases in California, 24% of all cases nationally despite having 12% of the U.S. population [4, 5]. Of Californian TB cases, 84% occur among persons born outside the United States and rates of tuberculosis among non-US-born Asians are 50 times that among US-born whites [6].”

Not sure why looking at attitudes/feelings around discrimination – while interesting, they don’t really add to the efforts/next steps around TB prevention (or this wasn’t made clear in the paper)

Revised manuscript to underscore the connection between racial discrimination and TB prevention activities. “Barriers to healthcare access and utilization, such as cost and racial/ethnic discrimination, affected less than one in ten persons. Racial or ethnic discrimination has been shown to reduce access care and filling of prescriptions which would hamper TB prevention activities [26, 27].”

Line 230 – not clear what % of homeless are foreign born. They couldn’t be surveyed, so how do we know they are not a large %?

Important point. We added a section in the limitations to address this issue: “Because this analysis focuses on non-U.S.-born persons and persons experiencing homelessness maybe under-represented, it would be helpful to know the proportion of non-U.S.-born persons who experience homelessness in California. Unfortunately, there is no statewide estimate for non-U.S.-born homelessness, but a recent study found the proportion of lifetime homelessness among non-U.S.-born persons was 1% [28].”

Need to clearly elucidate next steps with regard to improving EMR documentation of country of birth. Is there anything that can be included in the next statewide survey to help this TB campaign?

Agreed. Added this language to the discussion of EMRs: “EMRs should be modified to accommodate country of birth data and workflows should be modified to collect these data, without which a large portion of the at-risk population could be missed.”

We continue to work with partners to include TB questions on the statewide survey though to date questions on TB have not been included.

Good point about leveraging comorbidity work (smoking/diabetes).

Agreed. Helpful to know this made sense to the reviewer.

It seems like different countries at birth (e.g. Mexico) have different needs – maybe focus on some of those. County of birth being Mexico has the largest proportion of TB cases (in Table 1, 21), and in lines 230-231 you state ¼ are uninsured – seems like a good part to focus prevention efforts on. This pop also has a high percentage of diabetes.

We have identified several populations that require increase testing and treatment. From the conclusion: “We envision these efforts will be multi-faceted, multi-sectorial, and adapted to specific communities and providers and that they include input from populations at risk for TB.” 

Focus on provider education in terms of increased testing for TB in this population/routine testing (not patient – in line 232). And who are these providers caring for foreign born populations? (do they differ than other providers?)

We believe that both patient and provider awareness is important to successful intervention to increase testing and treatment. Provider engagement is discussed in the context of co-morbidities: “The prevalence of these risk factors among populations already at risk for TB provides opportunities for engagement with providers who may not currently think about TB even when seeing persons at risk for the disease. One engagement strategy could be using a common comorbidity such as diabetes as an entry point for provider education about TB.”

With regard to details on the providers who serve the population at risk, we are limited by this survey questionnaire which does not include much data on providers beyond language used at visit and the patient’s usual source of care.

What about the following factors in describing your pop of interest: legal status (esp for people interested in obtaining legal status – might be less likely to utilize any services out of fear of jeopardizing their future greencards), travel back to country of origin (perhaps this is where they get TB, thru travel, and not just coming into the US with it), age at immigration to US, household size/make up of extended family (and if they have travel back to at-risk countries, or issues of household crowding making it easier to spread TB).

We are limited by the survey questionnaire which does not include data on travel to country of origin.

Would be nice know breakdown of active TB cases (foreign born vs. US born), does this follow the same patterns as described in this paper?

Added the following to the introduction: “California bears a disproportionate share of the TB burden in the U.S. In 2019, there were 2,113 TB cases in California, 24% of all cases nationally despite having 12% of the U.S. population [4, 5]. Of Californian TB cases, 84% occur among persons born outside the United States and rates of tuberculosis among non-US-born Asians are 50 times that among US-born whites [6].”

Table 1 – more “TB Burden” to be at top of table, not at bottom

Revised per comment.

Table 4 – include percentages

Other reviewers also commented on this table. I have revised the table to be more understandable. Percentages are included.

Reviewer #2: Overall, I found this an interesting analysis of the demographic, health, and healthcare seeking characteristics of the non-US born population in California. While this population is certainly important to identify, describe, and understand for TB control, the paper lacks a clearly presented conceptual framework that describes how these demographic and clinical characteristics: 1) impact TB care and outcomes; and 2) are unique to TB. 

Re-wrote introduction to address lack of conceptual framework and to describe how healthcare utilization is important in planning TB prevention activities. Essentially, TB prevention activities rely on access and use of healthcare. One of the main interventions is to have persons at-risk for TB be tested by their doctor and treated if needed.

The non-USB population is at risk for a variety of conditions (Hepatitis, COViD, e.g.) making it important to clearly describe the context for focus on non-USB more explicitly in the introduction. 

Add context for the focus on non-USB in introduction: “California bears a disproportionate share of the TB burden in the U.S. In 2019, there were 2,113 TB cases in California, 24% of all cases nationally despite having 12% of the U.S. population [4, 5]. Of Californian TB cases, 84% occur among persons born outside the United States and rates of tuberculosis among non-US-born Asians are 50 times that among US-born whites [6].”

A conceptual framework for why each of these data fields from CHIS were chose would also be helpful, especially for readers not as familiar with the domestic TB literature.

Revised to clarify connection between health utilization and tuberculosis : “To make progress against TB in California, the number of persons that are tested and treated for LTBI needs to be substantially increased. Recent mathematical models support the scale-up of targeted testing and treatment of non-U.S.-born persons for TB infection as important for the reduction of TB in California [8, 9]. However, these TB prevention activities rely on access and use of healthcare and our knowledge of these attributes among non-U.S.-born persons is incomplete. In California and elsewhere, public health programs are engaging with community groups especially those focused on healthcare and representing affected communities but more information is needed about the demographics, healthcare utilization and potential barriers to care of populations at-risk for TB [10, 11]. Recent studies have highlighted disparities in healthcare utilization among minority groups but have not focused on non-U.S.-born persons at-risk for TB [12, 13]. Understanding healthcare utilization of at-risk populations is important for the planning of TB prevention activities.

Please specify in Methods why you focus on 5 countries named (assuming based on country of origin for most TB cases in CA?).

We revised the methods section to address this comment: “We calculated survey proportions and 95% confidence intervals (CI) stratified by country of birth, focusing on the six countries with the highest number of TB cases in California, specifically Mexico, United States, Philippines, Vietnam, China, and India. These countries accounted for 80% of all cases in California during 2016-2017 [18, 19].” 

Suggest a citation for this statement as it forms a key underlying premise for the purpose of the paper (ln 67)” One specific barrier to implementing TB prevention in large health systems is that country of 68 birth, the most important risk factor for TB, is often not captured in the electronic medical 69 record (EMR).”

 Citation added.

This statement does not seem to make sense (ln 277-278) “Our results show that non-English language at medical visit was an excellent proxy for non-U.S. 278 birth, but identified only half of all non-U.S.-born persons.” How is a proxy excellent if it only identifies half of the eligible population? Please refer to the 2-way table you present in results as to how you would like the reader to interpret this statement. Are you say that the positive predictive value or specificity is high but sensitivity is low? If so, please state as such.

Agreed. We have revised the language toclarify: “Fifth, persons who used a language other than English at their medical visit were highly likely to be born outside the U.S. Thus, preferred language may be suitable as a starting point for identifying high-risk subgroups. However, non-English language at medical visit identified only half of all non-U.S.-born persons.”

We also mention high specificity and low sensitivity in the results section: “As a proxy for nativity, language at medical visit had a low sensitivity (42%) and a high specificity (99%) (table 3).” Please note the table numbering has changed for tables 3 and 4.

Please describe important limitations of using CHIS data, ie, sampling bias based on phone and self-response and how this may actually underestimate the prevalence of some of those key sociodemographic determinants that affect TB care.

Agreed. The following text was added to the limitation section: “The survey was conducted by phone and may under-represent persons who are less likely to answer the phone, such as persons experiencing homelessness, young adults, and non-English speakers. Furthermore, it relies on participant self-report which may underestimate key determinants of TB risk, such as poverty and country of origin, because of social desirability bias.” 

Table 3 is challenging to read and interpret. Suggest revising.

Agreed. We have simplified the table by splitting it into two parts and relegating one part to the supplement. The table is now table 4 and Supplement Table 1.

You may want to consider using the first paragraph of the Discussion as the last paragraph of the Introduction, as it is the first place that seems to clearly articulate the purpose for the analysis presented.

Yes, very helpful suggestion. We have moved the first paragraph of the discussion to the end of the introduction.

Reviewer #3: 

This is an interesting paper. It has the potential benefits in the field of TB in the USA. But to let it be strong enough, some clarifications are needed. Below are some suggestions which could help improve the paper if the authors found them relevant.

- I would suggest to further explain the methodology. I’m a little bit concerned with the methodology section as presented. The methods section should be clearly and unambiguously stated. I find that there are many references that are indicated and that unfortunately hide a lot of details. For example, the methodology does not show the countries or let's say the different nationalities (India, Mexico, USA, Philippine, Vietnam, etc.) mentioned in the results section. Is there any link or relationship between individuals regarding their nationalities and the risk to get TB? Also, what is the threshold and the criteria of poverty according to the place of birth? According to the fact that poverty in the US could be completely different of the one in Philippine or somewhere else…

We have revised the methods section to address these concerns. With regard to countries of origin we have included the following text: “We calculated survey proportions and 95% confidence intervals (CI) stratified by country of birth, focusing on the six countries with the highest number of TB cases in California, specifically Mexico, United States, Philippines, Vietnam, China, and India. These countries accounted for 62% of all cases in California during 2016-2017 [14, 15].”

In reference to poverty, we added this language: “Poverty was defined as less than 139% of the U.S. federal poverty level. The respondent’s percentage of federal poverty level was calculated using household income earned in the U.S. and household size.”

- HMO. I don’t know what this acronym means! Please put the definition in bracket (p.99)

Apologies for this oversight! We have included the full wording now in the methods section: “Usual source of care was condensed from seven categories to three as follows: doctor’s office, health maintenance organization (HMO) or Kaiser were grouped as doctor’s office…”

- In most of the items presented in the results section, it is not easy to perceive the link between the different factors (diabetes, smoking, good or bad health, etc.) mentioned and their relationship with TB. The authors should discuss this further.

Revised the discussion to clarify the link between risk factors and tuberculosis. Here’s the passage from the discussion section: “Third, risk factors for the progression of latent TB infection to TB disease varied widely among the populations examined. Diabetes and smoking are associated with increased risk of progression of latent TB infection to active TB disease [23, 24]. Self-reported diabetes among Philippines-born and Mexican-born persons was more than double the rates for persons born in other countries including the U.S., similar to prior work [25, 26]. Likewise, former smoking was twice as high among Philippines-born and Mexican-born persons than among persons born in China, India or Vietnam as has been previously documented [27, 28]. The prevalence of these risk factors among populations already at risk for TB provides opportunities for engagement with providers who may not currently think about TB even when seeing persons at risk for the disease.” We also add the following text to the methods section “Smoking and diabetes were considered because they are risk factors for progression of latent TB infection to TB disease [20, 21].”

- The discussion section, as presented, in my opinion is more a part where the authors interpret their results and not discussing them. The discussion, as its name indicates, should be the place where the results, while being interpreted, are also confronted with the studies of other authors and where the points of view are clearly confirmed, refuted or nuanced. And this should be felt in the richness of the references mentioned in the said section. But, as we can see here, the authors have mentioned only 2 references in this whole long discussion... And besides, the article is not rich enough in terms of bibliographic references. So, I suggest to enrich it further, if only with a little more reading...

We a revised the discussion to juxtapose our results with results from the literature and have now included nine references in this section.

- Where is the conclusion section? And what are the main conclusions of the data obtained from the California Health Interview survey (CHIS)?

We have added a conclusion section: “Our analysis points out several important differences in demography, healthcare access and utilization, and language use among populations at risk for TB by country of birth. These differences present opportunities and signal potential pitfalls for planning outreach to specific communities at risk for TB and the providers, clinic systems, health insurers and health insurance purchasers who serve them.”

- What could be the concrete added value of this article? What contribution and especially how can the results obtained contribute to the elimination of TB in the whole USA in general or at least in California more specifically? This is not clear in the paper.

We have include extra language on the concrete value added of this article in the conclusion: “These data could be used to inform public awareness campaigns, provider education, academic detailing, and community outreach efforts. We envision these efforts will be multi-faceted, multi-sectorial, and adapted to specific communities and providers and that they include input from populations at risk for TB. While the information here most directly supports this tailored approach in California, we believe that similarly crafted approaches would help elimination TB in other states and across the U.S.”

---

## [Decision Letter · Decision Letter 1]

1 Nov 2021

PONE-D-21-08860R1Demographics and healthcare utilization of populations at risk for tuberculosis, California 2014-2017PLOS ONE

Dear Dr. Readhead,

Thank you for submitting your manuscript to PLOS ONE. After careful consideration, we feel that it has merit but does not fully meet PLOS ONE’s publication criteria as it currently stands. Therefore, we invite you to submit a revised version of the manuscript that addresses the points raised during the review process.

We look forward to receiving your revised manuscript.

Kind regards,

Rubeena Zakar, Ph.D

Academic Editor

PLOS ONE

Journal Requirements:

Reviewers' comments:

Reviewer's Responses to Questions

**Comments to the Author**

1. If the authors have adequately addressed your comments raised in a previous round of review and you feel that this manuscript is now acceptable for publication, you may indicate that here to bypass the “Comments to the Author” section, enter your conflict of interest statement in the “Confidential to Editor” section, and submit your "Accept" recommendation.

Reviewer #3: All comments have been addressed

Reviewer #4: (No Response)

Reviewer #5: All comments have been addressed

2. Is the manuscript technically sound, and do the data support the conclusions?

Reviewer #3: Yes

Reviewer #4: Yes

Reviewer #5: Yes

3. Has the statistical analysis been performed appropriately and rigorously? 

Reviewer #3: Yes

Reviewer #4: Yes

Reviewer #5: Yes

4. Have the authors made all data underlying the findings in their manuscript fully available?

Reviewer #3: Yes

Reviewer #4: Yes

Reviewer #5: Yes

5. Is the manuscript presented in an intelligible fashion and written in standard English?

Reviewer #3: Yes

Reviewer #4: Yes

Reviewer #5: Yes

6. Review Comments to the Author

Reviewer #3: The authors made a great effort to take into account all the comments and observations made in my previous review!

Reviewer #4: All comments are included in the attachment review report, the authors are particularly recommended to give adequate attention revising the manuscript for coherence of ideas and some areas need further clarification. For instance abstract.

Reviewer #5: Thank you for addressing the points highlighted in the previous review. The manuscript is sound and meets the criteria to be published.

7. PLOS authors have the option to publish the peer review history of their article (what does this mean?). If published, this will include your full peer review and any attached files.

Reviewer #3: **Yes: **Dr Nourou Barry

Reviewer #4: No

Reviewer #5: No

---

## [Author Response · Author response to Decision Letter 1]

19 Nov 2021

Second Revision - Response to Reviewers:

Reviewer #4: All comments are included in the attachment review report, the authors are particularly recommended to give adequate attention revising the manuscript for coherence of ideas and some areas need further clarification. For instance abstract.

We have complete extensive revisions to improve the coherence of ideas and clarify areas of the manuscript that were muddled, especially the abstract. I have included the point by point comments and responses here, but it may be more convenient for the reviewer to view them in context in the track changes version of the manuscript which has been uploaded as part of this submission.

Title: “unncessary authors can discuss about demographics under the health care utilization title-----"

Revised per request. Modified text to avoid stigmatizing language

Title: “I have searched to know who are popln at risk but nowhere the authors deined “

Very good point. I have included the definition in the methods section (line 109) and revised the introduction to highlight these groups (line 58).

Comment on short title: “not sure whether this is important”

Submissions require short title under 100 characters

Comment on abstract methods: “very much shallow make it has to be detail”

Made revisions here to provide more detail on the methods.

Comment on abstract methods language which recapitulated old mansucript title: “not clear how relevant”

Deleted reference to language use here. The relevance is discussed in the manuscript

Comment on abstract conclusion: “better conculusion”

Revised the conclusion to highlight an important finding which was that populations who experience risk for TB had health insurance and used healthcare. 

Line 100: “why?”

Modified language to explain why adult survey was used.

Methods, Line 104: “very large pargraph better to make short and clear paragraphs”

moved to appropriate subsection below.

Line it is better if the authors descibe each methodology scetion separately

Agreed. Added subsection headers and re-arranged text to fit into subsections. 

Methods: “this seems operational definition of terms,I think authors need to have operational definition section”

Agreed. Add subsection headers and re-arranged text to fit into subsections.

Results: “I think authors are not following plos one author guide for writing tables and graphs”

I have reviewed the table requirements and I believe that all tables in this document comply with the author guidelines. Can the reviewer or editor please specify what about the tables does not follow journal style guidelines?

Results section, health insurance subsection: “how do authors corrplate health insurance with utilization of health service;healthcare financing vs utilization. is it this is not out of the scope of this study”

The focus of the paper is on the differences in health utilization between populations who experience risk for TB. Health insurance is a key component of healthcare access, but alone it does not ensure utilization. That is why we looked at both health insurance and healthcare utilization. Again, our interest was to ¬understand if these populations had different insurance or utilization levels.

Results section, Overall Health and Risk factors for TB reactivation subsection: “journal requirment”

As above, apologies, I am unclear as to what does not meet the journal style requirement here. Can you specify what I can do to bring this in to compliance with journal style guidelines?

Table 3: “such descriptions are expected in testing new tests and gold standard test in epidemiology why authors included here, not clear”

In the introduction, we discussed that country of birth, a key risk factor for TB, was missing in many EHRs but that preferred language was used as a proxy. Here, we are setting up results for the use of language at medical visit as a method for identifying persons born outside the U.S. We thought that the standard calculations on sensitivity and specificity would be help readers understand how language could be used to identify non-U.S.-born. We draw conclusions from these results in the discussion (line 341). We have added language in discussion to further clarify the use of language as a proxy. 

Discussion: “what will be the recommendation of the authors for this result;aganist health insurance”

Apologies, I’m not sure I fully understand the comment. The result that most persons who experienced risk for TB had health insurance is a positive one, in that, it suggests that having health insurance is not a barrier to accessing healthcare for many in these populations. We highlight the importance in line 291 “With the notable exceptions of persons experiencing homelessness and one quarter of persons born in Mexico who are uninsured, the hurdle of getting at-risk populations into care has largely been met.” We are certainly not recommending against health insurance. Perhaps the reviewer could further explain?

Acknowledgements: “many of the jouranl requrements are not fullfilled”

As above, apologies, but after reading through the journal requirements again, I am unsure as to what about the tables and table references is not in compliance with journal requirements. Could you specify what in particular about the tables and table references does not meet journal requirements?

---

## [Decision Letter · Decision Letter 2]

13 Jan 2022

PONE-D-21-08860R2Health insurance, healthcare utilization and language use among populations who experience risk for tuberculosis, California 2014-2017.PLOS ONE

Dear Dr. Readhead,

Thank you for submitting your manuscript to PLOS ONE. After careful consideration, we feel that it has merit but does not fully meet PLOS ONE’s publication criteria as it currently stands. Therefore, we invite you to submit a revised version of the manuscript that addresses the points raised during the review process.

We look forward to receiving your revised manuscript.

Kind regards,

Rubeena Zakar, Ph.D

Academic Editor

PLOS ONE

Journal Requirements:

Reviewers' comments:

Reviewer's Responses to Questions

**Comments to the Author**

1. If the authors have adequately addressed your comments raised in a previous round of review and you feel that this manuscript is now acceptable for publication, you may indicate that here to bypass the “Comments to the Author” section, enter your conflict of interest statement in the “Confidential to Editor” section, and submit your "Accept" recommendation.

Reviewer #6: All comments have been addressed

Reviewer #7: (No Response)

Reviewer #8: (No Response)

2. Is the manuscript technically sound, and do the data support the conclusions?

Reviewer #6: Yes

Reviewer #7: Yes

Reviewer #8: Partly

3. Has the statistical analysis been performed appropriately and rigorously? 

Reviewer #6: (No Response)

Reviewer #7: Yes

Reviewer #8: No

4. Have the authors made all data underlying the findings in their manuscript fully available?

Reviewer #6: Yes

Reviewer #7: Yes

Reviewer #8: No

5. Is the manuscript presented in an intelligible fashion and written in standard English?

Reviewer #6: Yes

Reviewer #7: No

Reviewer #8: Yes

6. Review Comments to the Author

Reviewer #6: Tuberculosis (TB) is the disease that should be paid attention as one of the re-emerging infectious ones. The authors concluded that analysis of CHIS provided several important differences by country of birth and language for planning outreach, public awareness campaigns, and provider education etc.

This paper told the case of California, however, information from this would be useful in states with many emigrants for controlling the occurrence of TB and other infectious diseases.

The title should reflect the conclusion to some extent.

Reviewer #7: They presented well and followed most of the guidlines, but you need to improve standard english for a scientific research work.

Reviewer #8: General comments:

As the article defines being born overseas (in high TB risk countries) as the main risk factor for TB why are US-born people included (or are they the comparison group and if so why not also include other low TB countries)?

I though the aim of the paper was to identify how health insurance, healthcare utilization and language use impact on those at risk of TB and yet no comparison analysis was conducted (adjusting for confounders).

There is no description of the statistical methods used for the descriptive comparisons and it appears that no models were developed.

The tables on language at consultation and by country does not add much to the paper as apart from providing the % of the population who also speak English.

Because of the limited analysis, including no confounder adjustments, the conclusions are limited and not very informative.

In order for the article to be informative the aims, definitions and analysis needs to be reconsidered.

7. PLOS authors have the option to publish the peer review history of their article (what does this mean?). If published, this will include your full peer review and any attached files.

Reviewer #6: **Yes: **SHINICHI ARAKAWA

Reviewer #7: **Yes: **Abdulkadir ISMAEL Ahmed

Reviewer #8: **Yes: **A/Prof Margo Barr

---

## [Author Response · Author response to Decision Letter 2]

16 Mar 2022

Response to Reviewers:

Reviewer #6: Tuberculosis (TB) is the disease that should be paid attention as one of the re-emerging infectious ones. The authors concluded that analysis of CHIS provided several important differences by country of birth and language for planning outreach, public awareness campaigns, and provider education etc.

This paper told the case of California, however, information from this would be useful in states with many emigrants for controlling the occurrence of TB and other infectious diseases.

The title should reflect the conclusion to some extent.

Thanks for this comment. We believe that this analysis has implications for TB elimination in other states and may potentially be helpful for the control of other disease of public health interest. However, given that these data are focus solely on California and these populations where chosen because they experience a higher risk of TB, we think that the title as it stands reflects the scope of the data well.

Reviewer #7: They presented well and followed most of the guidlines, but you need to improve standard english for a scientific research work.

We have made changes throughout the manuscript to clarify language. We have added Oxford commas throughout and simplified language in multiple passages.

Reviewer #8: General comments:

As the article defines being born overseas (in high TB risk countries) as the main risk factor for TB why are US-born people included (or are they the comparison group and if so why not also include other low TB countries)?

We revised the methods section to address this question. The language is as follows:

“We chose to examine health insurance, healthcare utilization and language use among persons born in Mexico, the United States, the Philippines, Vietnam, China, and India. When stratifying by country of birth, these six countries had the highest number of TB cases in California. Together they accounted for 80% of all cases in California during 2016-2017. Thus, for the purposes of this analysis, we defined populations who experience risk of TB as persons born in these six countries.”

I though the aim of the paper was to identify how health insurance, healthcare utilization and language use impact on those at risk of TB and yet no comparison analysis was conducted (adjusting for confounders).

Thank you for this comment. It indicates that we have not adequately laid out the aim of this analysis. The aim is not to assess the impact of health insurance and other factors on the risk of TB but rather to describe health insurance and other factors such that we can better plan TB interventions. With this in mind, we revised the final paragraph of the introduction to read:

“Our aim was to describe the health insurance, healthcare utilization and language use of groups that experience risk for TB in California using a large-scale, population-based health survey to inform planning of TB prevention activities. As California transitions from TB control activities focused on finding and treating active TB disease to TB prevention activities requiring latent tuberculosis infection (LTBI) testing and treatment among populations who experience risk for TB, detailed knowledge about the population of the ten million persons in California born outside the U.S. is crucial.”

There is no description of the statistical methods used for the descriptive comparisons and it appears that no models were developed.

The statistical methods for the descriptive comparisons are laid out in the Method section under statistical analysis. I have provided an excerpt here.

“We calculated survey proportions and 95% confidence intervals (CI) stratified by country of birth. We adjusted the following variables by age: doctor’s visit in the last 12 months, overall health, smoking and diabetes. We used the 2017 U.S. Census annual estimate of population for California as the standard population [22].”

With regard to the development of models, we believe that descriptive analysis is sufficient to complete the main aim. We were interested in understanding the health insurance coverage, health utilization and language use of these populations which have a fairly high risk of TB such that public health programs can plan to engage these populations more efficiently. For example, if programs were planning printed flyers to engage persons born in the Philippines, they might reasonably use some flyers printed in English and print a limited number of flyers in Tagalog based on the fact that 93% speak English well, very well or only speak English. In contrast, when planning flyers for persons born in Vietnam, they might reasonably print more flyers in Vietnamese as 53% of this population speaks English not well or not at all. We outline this point in our conclusion excerpted below:

“Our analysis points out several important differences by country of birth in demography, healthcare utilization, and language use among populations who experience risk for TB. These differences present opportunities and signal potential pitfalls for planning outreach to specific communities that experience risk for TB and the providers, clinic systems, health insurers and health insurance purchasers who serve them. These data could be used to inform public awareness campaigns, provider education, academic detailing, and community outreach efforts.”

The tables on language at consultation and by country does not add much to the paper as apart from providing the % of the population who also speak English.

 We outlined the use of language at medical visit as potentially a reasonable substitute for the main risk factor for TB in the U.S., country of birth, which is often missing in medical records. From the point of view of a public health program considering ways to increase appropriate TB testing, we view this result as one of the most useful in the manuscript.

“One specific barrier to implementing TB prevention is that country of birth, the most important risk factor for TB, is often not captured in the electronic medical record (EMR) [16]. Without country of birth data, technology available in the EMR, such as prompts or reflex testing, cannot be used to promote TB testing among these at-risk groups. Preferred language at medical visit, which is often populated in EMR data, may be useful as a proxy for country of birth.” 

We also argue for the usefulness of our results in the discussion:

“Fifth, we investigated whether language at medical visit could be used to identify persons born outside the U.S. We found that persons who used a language other than English at their medical visit were highly likely to be born outside the U.S. Thus, preferred language may be suitable as a starting point for identifying high-risk subgroups.”

Because of the limited analysis, including no confounder adjustments, the conclusions are limited and not very informative.

In order for the article to be informative the aims, definitions and analysis needs to be reconsidered.

It is true that we did not build multivariable models or make adjustments for confounders. The aims of this analysis were to describe health insurance, health utilization and language use among populations that experience risk for TB. These pieces of information are crucial to progressing toward TB elimination. From the conclusion: 

“These differences present opportunities and signal potential pitfalls for planning outreach to specific communities that experience risk for TB and the providers, clinic systems, health insurers and health insurance purchasers who serve them. These data could be used to inform public awareness campaigns, provider education, academic detailing, and community outreach efforts. 

Our aim was not to predict an outcome or to provide evidence of causation between an exposure and outcome. Thus, we did not use multivariable models. But considering the aim of the analysis (which we have clarified in responses above), those methods did not seem to us to fit the job. The conclusions are simple but we do not think they are limited. One quarter of California’s population was born outside the United States. That’s 10 million people. If we can understand the health insurance coverage, health utilization and language use of persons born in these six countries, we have started to understand how we can engage these populations and hopefully bring some TB and other diseases that affect some of these populations.

---

## [Decision Letter · Decision Letter 3]

9 May 2022

Health insurance, healthcare utilization and language use among populations who experience risk for tuberculosis, California 2014-2017.

PONE-D-21-08860R3

Dear Dr. Readhead,

We’re pleased to inform you that your manuscript has been judged scientifically suitable for publication and will be formally accepted for publication once it meets all outstanding technical requirements.

Kind regards,

Rubeena Zakar, Ph.D

Section Editor

PLOS ONE

Additional Editor Comments (optional):

Reviewers' comments:

Reviewer's Responses to Questions

**Comments to the Author**

1. If the authors have adequately addressed your comments raised in a previous round of review and you feel that this manuscript is now acceptable for publication, you may indicate that here to bypass the “Comments to the Author” section, enter your conflict of interest statement in the “Confidential to Editor” section, and submit your "Accept" recommendation.

Reviewer #6: All comments have been addressed

Reviewer #7: All comments have been addressed

2. Is the manuscript technically sound, and do the data support the conclusions?

Reviewer #6: Yes

Reviewer #7: Yes

3. Has the statistical analysis been performed appropriately and rigorously? 

Reviewer #6: Yes

Reviewer #7: Yes

4. Have the authors made all data underlying the findings in their manuscript fully available?

Reviewer #6: Yes

Reviewer #7: Yes

5. Is the manuscript presented in an intelligible fashion and written in standard English?

Reviewer #6: Yes

Reviewer #7: Yes

6. Review Comments to the Author

Reviewer #6: The manuscript has been adequately revised.

The content of this study is interesting, and the information obtained from this would be reflect 険the health administration in California and also in the United States.

Reviewer #7: I see that most of previous reviews whether my comments or that from other reviewers' were addressed.

7. PLOS authors have the option to publish the peer review history of their article (what does this mean?). If published, this will include your full peer review and any attached files.

Reviewer #6: **Yes: **Shinichi Arakawa

Reviewer #7: **Yes: **Abdulkadir ISMAEL Ahmed

---

## [Editor Report · Acceptance letter]

16 May 2022

PONE-D-21-08860R3 

Health insurance, healthcare utilization and language use among populations who experience risk for tuberculosis, California 2014-2017. 

Dear Dr. Readhead:

I'm pleased to inform you that your manuscript has been deemed suitable for publication in PLOS ONE. Congratulations! Your manuscript is now with our production department. 

Kind regards, 

on behalf of

Dr. Rubeena Zakar 

Section Editor

PLOS ONE